# Absorption Performances of PLA-Montmorillonite Nanocomposites Thin Films in Salisbury and Rozanov Configurations: Influence of Aging and Mechanical Recycling

**DOI:** 10.3390/mi13122152

**Published:** 2022-12-05

**Authors:** Lakhdar Sidi Salah, Nassira Ouslimani, Yann Danlée, Isabelle Huynen

**Affiliations:** 1Research Unit Materials, Processes and Environment (URMPE), Faculty of Technology, Université M’Hamed Bougara Boumerdès, Boumerdes 35000, Algeria; 2Processing and Shaping of Fibrous Polymers Laboratory, Faculty of Technology, Université M’Hamed Bougara Boumerdès, Avenue of Independence, Boumerdes 35000, Algeria; 3Institute of Information and Communication Technologies, Electronics and Applied Mathematics (ICTEAM), Université Catholique de Louvain, Place du Levant 3, 1348 Louvain-la-Neuve, Belgium

**Keywords:** PLA composites, organo-modified montmorillonite, EMI shielding, artificial aging, mechanical recycling, microwave absorption, Salisbury, Rozanov

## Abstract

The present paper aims to address the crucial concern of pollution induced by growing plastic waste and electromagnetic interference (EMI). Nanocomposites combining poly(lactic acid) (PLA) and organo-modified montmorillonite (OMMT) are synthesized and compression molded into thin films. A first set of samples, referred as virgin, was kept as is, while a second set of samples were photochemically, thermally and hydrolytically aged before mechanical recycling via extruding and second compression molding, resulting in the so-called recycled composite. The electromagnetic (EM) properties with a focus on microwave absorption performances of virgin and recycled samples are compared for various thicknesses and weight concentrations of OMMT in PLA matrix. The EM performances are gauges by Rozanov and Salisbury structures that consist in one- and two-layer stacks of composite films back-coated by a metal foil. Characterization in Rozanov configuration shows an average absorption index over the Ka band of 29.3% and 21.1% for, respectively, virgin and recycled PLA reinforced with 4 wt.% OMMT. An optimization of the film thickness is proposed; up to 61.85% and 80% of absorption with a thickness of 1.4 mm and 3.75 mm, respectively, is reached with a metal back-coated rPLA-4%OMMT film. Characterization in Salisbury configuration gives advantage to the recycled structure with an average absorption of 49.6% for a total thickness of 1.4 mm. The requirements of EMI shielding are met by PLA-OMMT composites with a certain benefit of recycling process on EM performance.

## 1. Introduction

There is a growing need for lightweight devices combining compactness and reliability, due to the increase in communications systems [1]. The constraints are antagonist if we consider that reduced size increases the risk of harmful electromagnetic interference between electronic components and the end-user [2]. Research is conducted for more than one decade on microwave absorbing structures (MASs) designed to prevent such interference [3,4,5]. Among these, polymer composites combining conductive charges and polymer hosting matrix offer several advantages; high conductivity at low loading rate owing to very good dispersion, mechanical stiffness as well as lightweight compared to metallic shields, conformability for stealth application, etc. [6,7]. MAS can be designed to reduce the radar cross-section (RCS) of various objects for military as well as civil applications: an absorbing layer on the target reduces the reflected signal, making the target stealth. Well-known applications are military airplanes (stealth bombers) and towers of wind farms, that need to be stealth for security reasons, i.e., to ensure invisibility or avoid spurious signatures perturbing the civil air traffic [8]. The Salisbury screen configuration [9,10], is developed to reduce the RCS. It consists of an absorbing layer covering the target metallic assumed to be reflective. A well-known case is radar communications at X band (8–12.4 GHz) [11]. Various MAS have been studied where carbon fillers, including carbon black, incorporated at different concentration levels in single as well as multilayer configurations, induce reflection below 10 dB [12,13]. An efficient absorber must minimize the reflection at its surface, while maximizing absorption over its thickness. Conductive composites are suited to attenuate the signal; however, conductive charges may also increase the reflection, as do metallic shields. An adequate strategy must be set up in order to counterbalance this effect [8]. Most research is devoted to surface modification and high concentration of multi-fillers in nanocomposites, as well as automated numerical techniques to synthesize randomly arranged multilayer topologies of nanocomposites [14,15,16,17,18].

On the other hand, sustainability is becoming a major challenge for the coming decade. The European Parliament promotes actions in favor of circular economy of materials, aiming to reduce the world’s plastic growing amount of plastic waste [19,20]. To address this issue, several alternatives for the valorization of plastic waste have been proposed and evaluated, such as mechanical recycling [21].

Among all the plastics used in the industry, PLA arouses interest thanks to its “green” side [22]. PLA is obtained from renewable raw materials and releases almost no toxic gases or CO${}_{2}$ emissions. It is a polyester resulting from the fermentation of a source of carbohydrates such as sugar cane or corn starch [22]. Nevertheless, the recyclability of PLA remains a challenge for today’s industry [23]. Even though this material is recyclable, there are only a few facilities that support it; currently, a significant portion ends up in landfill. An intelligent valorization of this polymer therefore arouses interest for a more environmentally responsible society.

In this context, the present paper investigates the effect of aging and mechanical recycling on the performances of poly(lactic acid) (PLA)-clay nanocomposites on the electromagnetic performances including stealth capacity in Salisbury configuration, where the composite sample is backed by a metallic plate. Section 2 exposes the fabrication process of the composites and the experimental measurement setup, while Section 3.1 and Section 3.2 expose the theoretical concepts associated to Rozanov and Salisbury configurations, respectively. In Section 4.1, the electromagnetic properties of virgin and recycled PLA-based nanocomposites are studied. Section 4.2 compares the reflectivity and absorption performances of both families of composites in Rozanov configuration, and Section 4.3 optimizes the absorption performance via a tuning of the film thickness. Section 4.4 compares the reflectivity and absorption performances of both families of composites in Salisbury configuration. Summary and perspectives are given as a conclusion of this work.

## 2. Materials and Methods

### 2.1. Fabrication of Nanocomposite Samples

Neat PLA pellets from NatureWorks were processed by melt extrusion in a twin-screw microcompounder (L/D = 20) at 60 rpm and a residence time of 2.5 min. The temperature profile, from hopper to die was: 125–165–190–190–180 °C. The obtained material was then hot pressed with these following conditions: (I) 4 min at 190 °C, no pressure, to melt the samples, (II) 1 min at 190 °C under 15 MPa, and (III) 5 min between cooling plates at 25 °C under 15 MPa. A set of films with a thickness of 200 and 400 µm are thereby produced. The films of virgin PLA (VPLA) were then subjected to an accelerated ageing protocol including: (I) 40 h of photochemical ageing in the Atlas UVcon™ chamber where samples were exposed to the radiation of eight F40UVB lamps for 40 h at room temperature, (II) 480 h of thermal ageing in an oven at 50 °C and (III) 240 h of hydrolytic ageing in deionized water at 30 °C. The aged samples were then washed at 85 °C in a NaOH (1 wt.%) and Triton X (0.3 wt.%) for 15 min, dried, grinded and reprocessed by melt extrusion and compression molding, resulting in recycled PLA samples (rPLA). Addition of organically modified montmorillonite (OMMT), Cloisite^®^ 30B Nanoclay from Southern Clay Products was performed in a second melt extrusion process including either virgin or recycled PLA, resulting in virgin and recycled PLA–OMMT nanocomposites. Only PLA undergoes the process of ageing, the addition of fresh OMMT is made in the second step of mechanical recycling for recycled PLA nanocomposites.

### 2.2. Microwave Characterization

The electromagnetic characterization was performed over the Ka band (frequency range between 26.5 and 40 GHz) using an Anritsu M54644B Vector Network Analyzer (VNA) in wave guide configuration as described in ref. [24]. The calibration is made by LRL/LRM method, and the IF bandwidth is set at 300 Hz. The VNA measures at its two access ports, noted 1 and 2, the fraction of microwave power that is reflected by the composite sample, noted $S_{11}$ for port 1, and $S_{22}$ for port 2, as well as the fraction of power that is transmitted from port 1 to 2 ($S_{21}$) and reversely from port 2 to 1 ($S_{12}$). Figure 1 illustrates these S-parameters with the 2-port configuration over a material under test (MUT). From the measurement of $S_{11}$, $S_{22}$, $S_{21}$ and $S_{12}$, the microwave reflectivity and absorption in reflective radar Salisbury configuration are obtained, as will be discussed in Section 3.2.

## 3. Theoretical Concepts

### 3.1. Assessment of Performances Using Rozanov Formalism

The designs of radar absorbing materials (RAMs) often assume the presence of a metallic structure in concrete applications, which implies serious reflectivity. Removing reflection inevitably requires a thick coating for the absorption of longer wavelength radiation. Attempts to reduce the thickness of the absorber often result in a decrease in the bandwidth of good absorption, proportional to the thickness reduction achieved [25]. The performances of RAM can be gauged using the Rozanov formalism that allows us to predict the optimal theoretical thickness of a composite slab given its level of reflectivity and operation wavelength range. The Rozanov configuration as well as the Salisbury are back-coated by a perfect electrical conductor (PEC); only the $S_{1}1$ parameter is measured corresponding to the reflection through the structure. Figure 2 shows the standard configuration of the Rozanov configuration.

The Rozanov formula is given in ref. [27] as:(1)$$\int_{\lambda_{i}}^{\lambda_{f}}ln\left|R\left(\lambda \right)\right|\;d\lambda ⩽2\pi^{2}d_{R}$$
where $\lambda_{i}$ and $\lambda_{f}$ are the wavelength limits of the Ka band, and $d_{R}$ as the optimal thickness limit is finally given by: (2)$$d_{R}⩾\int_{\lambda_{i}}^{\lambda_{f}}\frac{ln|R(\lambda )|}{2\pi^{2}}\;d\lambda$$

The ultimate thickness $d_{R}$ gives the theoretical limit of any RAS including stack of homogeneous layers being potentially ferromagnetic. The gap between this ultimate $d_{R}$ and the measured thickness allows gauging the efficiency of the RAM.

### 3.2. Microwave Absorption in Salisbury Configuration

The microwave absorption capability of our samples is studied via the reflection induced by the sample in a Salisbury screen configuration, where the absorbing sample is backed by a metallic plate supposed to be a PEC. This configuration illustrated in Figure 3 mimics the radar operation where signal incident to a target is reflected, allowing its detection by the radar. This resonant structure is designed such as the thickness of the dielectric layer is equal to $\frac{\lambda }{4}$ of the wavelength [28]. As the wave is reflected because of the metal back coat, it is a half wavelength out of phase with the incoming wave. It results in a destructive interference rising the global absorption. The Salisbury configuration is adequate to evaluate the absorption capability of a material through the measurement of R. It means that all signal must be absorbed by the sample so that almost nothing is reflected (nor transmitted since the metallic plate blocks the propagation). For stealth applications, reflectivity R has to be minimized in order to make the target invisible. However, this Salisbury design maximizes the efficiency at a single radar frequency, consequently the enemy simply shift its radar frequency to defeat the stealth [25]. More complicated multilayer Salisbury designs, such as Jaumann absorbers, cover a band of frequencies, but the thickness is generally higher [25,28].

Reflectivity R in Salisbury configuration expresses as a function of the scattering parameters $S_{11}$ and $S_{21}$, that are measured for a composite slab using a Vector Network Analyzer (VNA) in a waveguide configuration described in ref. [29].
(3)$$R=S_{11}-\frac{S_{21}^{2}}{1+S_{11}}$$

The fraction of power that is reflected at input of the slab is proportional to *R*:(4)$$P_{R}={\left|R\right|}^{2}$$

Due to the presence of the back metal plate in Salisbury configuration, no power is transmitted, and the power balance reduces to:(5)$$P_{R}+P_{A}=1\to P_{A}=1-P_{R}$$
where $P_{A}$ is the absorbed power. Additional information is available in Appendix A. The Salisbury screens obtain a successive peak of absorption depending on the thickness of the lossy material, since destructive interferences occur. The frequencies of absorption are given by the equation:(6)$$f=\left(\frac{1}{4}+\frac{n}{2}\right).\frac{c}{t\sqrt{\varepsilon .\mu }}$$
where n∈N, *c* is the speed of light in vacuum, *t* is the thickness of the lossy material, and ε and μ are permittivity and permeability of the lossy material, respectively. As an example, the Figure 4 shows a simulation based on an analytical model of the chain matrix of the absorption index over a large spectrum for recycled PLA (lossy material) front-coated by PLAr–4%OMMT (resistive sheet). Below ∼30 GHz, the reflection prevents absorption while successive absorption peaks appear for higher frequencies.

The analysis concentrates on the thickness of the samples only, since the topologies of Figure 2 and Figure 3 assume an infinite extension over the surface of the sample; both simulations and measurement extractions do not require the knowledge of the surface of the sample; only its thickness is involved in equations given in Appendix A. The thickness is used as tuning parameter for the optimization of the absorption in Section 4.3. Let us note that all films are more than one order of magnitude above the skin depth, ultra-thin films would have different properties.

## 4. Results and Discussion

### 4.1. Electromagnetic Properties

Figure 5 shows measured real and imaginary parts of permittivity versus frequency for virgin and recycled polymer (rPLA) loaded with various wt.% of OMMT and with various thicknesses of processed samples. It is observed that for virgin samples (left-handed columns in Figure 5) both real and imaginary parts of permittivity increase with concentration in OMMT. This is in accordance with the Maxwell-Wagner-Sillars polarization effect [30] occurring in composite materials loaded with conductive inclusions. This effect is modeled via a capacitance network connecting OMMT inclusions separated by PLA matrix. This formalism explains why the real part of the permittivity increases with wt.% and compared to the value of neat PLA. However, when the frequency increases, the capacitance tend to short-circuit so that permittivity returns to the value of neat PLA [31]. These observations are valid whatever the thickness of the composite sample, i.e., 200 µm or 400 µm.

Recycling significantly increases the imaginary part of permittivity, which is beneficial to the absorption. The aging process involves the formation of ionic and radical species, which create new interfacial polarization in the polymeric matrix [32]. Those large dipoles can interact with the applied electric field and thus increase the imaginary part of permittivity [33,34]. The consequence of this higher imaginary permittivity of the recycled PLA films is exposed via the EM performances explained in Section 4.2, Section 4.3, Section 4.4.

Figure 6 shows the Cole-Cole (CC) representation of the permittivity for each sample analyzed in Figure 5. The imaginary part noted $\varepsilon_{r}^{\prime \prime }$ is represented as a function of the real part $\varepsilon_{r}^{\prime }$ of the permittivity, the frequency being an implicit variable. For each sample, a series of circles or semicircles are observed. These are typical of the interfacial polarization effects occurring in the nanocomposite. Indeed, each interface between OMMT and neat PLA is the site of accumulation of polarized charges, while neat PLA separating conductive OMMT inclusions act as an electrical capacitor. The resulting distributed RC network is responsible for the formation of (semi-) circles in the Cole-Cole representation [35]. However, curves are flattened/squeezed as compared with perfect hemi-circles that are associated in the impedance spectroscopy theory to simple RC parallel circuits. As explained in refs. [36,37], the deformation of the curves with respect to pure RC behavior is associated to a spatial distribution of RC circuits having different time constants. These distributed circuits mimic the dispersive and disordered nature of the PLA or OMMT material, along with spatial inhomogeneity and frequency dependence on the conductivity of the nanocomposites.

The comparison of CC plots for virgin (left columns) and recycled (right columns) reveals that mechanical recycling induces more expanded semicircles. This means that recycling modifies the microscopic organization and dispersion of OMMT in neat PLA matrix, resulting in more disorder in the composite system that affects the interfacial polarization mechanism.

The presence of several circles has been observed in many works [38,39,40]; it is associated to the existence of multiple dielectric Debye dielectric relaxation phenomena in the samples, as well as interfacial polarization mechanisms active at the multiple interfaces between the inclusions and the hosting PLA medium.

### 4.2. Reflection and Absorption Performances in Rozanov Configuration

Following the Rozanov formalism, Figure 7 and Figure 8 show the measurement of reflected and absorbed power, respectively. Reflection is measured through the S-parameters and displayed in function of wavelength. Absorption index is computed using Equations (3)–(5), for virgin and recycled PLA–OMMT nanocomposites, as a function of frequency and for various wt.% concentration in OMMT. The peaks and valleys of reflection over the frequency band correspond to the multiples of the quarter of the wavelength, inducing a destructive and constructive interference between the incident and reflected wave from metal coat. As illustrated in Figure 4 through the absorption index, this phenomenon is expected for a given thickness (and dielectric properties) of the MUT [41]. A residual mismatch of the waveguide flanges used for the measurements explains also the light oscillations observed on the curves [42].

Table 1 summarizes the results via the reported mean values averaged over the 26.5–40 GHz frequency range; the right column gives the Rozanov thickness $d_{R}$ of the simulated RASs. The first observation is that the absorption index of recycled nanocomposites is poorly dependent on the OMMT concentration. Reflectivity of pure PLA-based films is slightly lower than recycled ones, which involves a better absorption. It means the VPLA–OMMT composites have better impedance matching with incoming microwave [43,44]. Regarding Figure 5, the imaginary part of pure composites is systematically low, conferring a good impedance matching and therefore providing a low reflectivity. On the other hand, the imaginary part of permittivity helps the absorption of microwave energy, but it requires higher thickness to become significant. The next section studies the thickness aspect for maximizing absorption index.

### 4.3. Optimization of Thickness in PEC Back-Coated Configuration

As a matter of fact, the poor absorption levels observed in Figure 7 and Figure 8 are not sufficient to qualify our films as efficient absorbers. The simplest way to improve the performances is an increase in thickness of the films [45]. Based on the measured S-parameters of each film, the behavior for various thicknesses is extrapolated and the absorption index is computed. The mathematics of this procedure are detailed in Appendix A. The composites are simulated with a thickness set at 1.4 mm (the thickness is selected for a comparison with Salisbury screens described in Section 4.4). Figure 9 and Figure 10 and Table 2 summarize the performances obtained when tailoring the thickness of each virgin and recycled film having 0, 2, or 4 wt.% OMMT concentration.

The simulations reveal that recycled PLA samples are much more absorbent, with associated reflection level -2dB (see values in Table 2). When the thickness of an absorbing material becomes close to the wavelength, the microwave can be attenuated with conductive matter (because of the imaginary part of permittivity). This absorbing phenomenon is predominant for recycled films, so that absorption index increases despite the lower impedance matching observed here above (see Table 1) for thin films. On the other hand, the reflectivity of pure PLA nanocomposites prevents high energy absorption, resulting in a lower absorption index.

The Rozanov formalism favors very thin layer absorbing over large frequency band; nevertheless, this optimization focuses on absorption index despite thickness. The loss in terms of Rozanov efficiency is the gain here in terms of absorption. Let us note that a stack of multilayer could improve the Rozanov performance but drastically increase the complexity of implementation, considered as less appropriate in a recycling process.

Focusing on EMI shielding applications, we fix a criterion of performance of 80% of average absorption over the Ka band. It results in the rPLA–4%OMMT composites reaching this condition with a thickness ⩾ 3754 µm; the other films require higher thickness. Figure 11 illustrates the absorption of this film.

### 4.4. Reflection and Absorption Performance in Salisbury Configuration

Salisbury screens have been assembled with PLA-based films. The stack of films is sandwiched between the waveguides with a metal foil inserted on the port 2 side, the method is described in ref. [24]. The lossy material is recycled or virgin PLA participating to trapping of the microwave. Following the Equation 6, the thickness of PLA was set at 1.2 mm insuring a first peak of absorption at ∼36 GHz knowing that average dielectric constant is 3–3.4 and permeability is 1. The resistive layer must be the most conductive film in order to confine microwaves between PEC and this front layer. Several configurations have been examined with recycled or virgin composites, or a combination of them. Table 3 sums up the 3 Salisbury structures studied in this work.

The Salisbury screens depicted in Table 3 were characterized over a 26.5–40 GHz spectrum. The reflectivity is displayed in function of the wavelength and the absorption index in function of the frequency. Figure 12, Figure 13 and Figure 14 show the virgin, recycled and hybrid Salisbury structures, respectively; the blue line is the measurement made by VNA and the dotted line is the simulation based on the material properties extracted from characterization (shown in Figure 5).

The characterization proves that the recycled Salisbury structures minimize the reflectivity, and consequently maximizes the absorption. Regarding Figure 5, the real and imaginary parts of permittivity of rPLA–4%OMMT are higher than VPLA–4%OMMT; the recycled composite film is therefore more adapted as a resistive sheet according to the Salisbury concept. One also observes that the rPLA improves the trapped wave attenuation particularly thanks to the higher imaginary part of the permittivity. The hybrid Salisbury screen shows intermediate performances; it undergoes reflectivity since the VPLA slab does not significantly attenuate the energy. Those results drive the potential of recycled materials in the EMI shielding application, since the recycling process increases performances. Our Salisbury structures are located somewhat below the performance of Rozanov-like simulations. The best candidate is rPLA–4%OMMT composites with an average absorption of 61.85% in Rozanov configuration. On the other hand, the recycled Salisbury screen has an average absorption of 49.6% with a total thickness of 1.4 mm. A higher reflection of −2.1 dB due to inadequate impedance matching prevents this Salisbury screen from obtaining better EMI performance; indeed the first layer involves a certain reflection by its own trapping wave concept [46,47]. Despite the smart structure, the Salisbury is also limited in absorption because of the low amount of OMMT. The nanofillers help to attenuate the microwave so that a full homogeneous film of OMMT is more effective. Salisbury screens remain a convincing alternative since the consumption of OMMT is 7 times lower. Indeed, the price and rareness of these nanofillers motivates the use of low-concentration composites. The saving in OMMT of Salisbury screens gives them a sustainable advantage.

## 5. Conclusions

Organo-modified montmorillonite (OMMT) were mixed by extrusion with polylactic acid (PLA) matrix. The pellets were then shaped in thin films. The “recycled” polymer was subjected to accelerated ageing before mechanical reprocessing via extrusion and compression molding. The comparison of electromagnetic properties and microwave absorption performances of virgin and recycled samples for various thicknesses of films and weight concentration in OMMT reveals that mechanical recycling does not significantly modify the electromagnetic performances of the nanocomposite, proving that recycled PLA offer promising perspectives in the field of plastic waste reduction in the electronic industry. Both virgin and recycled PLA-based composites ensure an absorption rate superior to 20% for a thickness of 400 µm in the Rozanov configuration. As this value is not compliant with the requirements of EMI shielding, an optimization of the film thickness is proposed. The metal back-coated rPLA–4%OMMT film is demonstrated to be efficient to ensure an average absorption rate of above 61.8% at a thickness of 1.4 mm thick, and 80% at 3.75 mm (precisely 3754 µm). Salisbury structures have demonstrated mean absorption up to 49.6% with recycled PLA composites. This alternative mainly uses rPLA without the necessity of a substantial amount of OMMT. Both configurations, Rozanov and Salisbury, prove the advantage of recycling on the EM performances, promoting a serious interest for this method that is sustainable for the environment.

## Figures and Tables

**Figure 1 micromachines-13-02152-f001:**
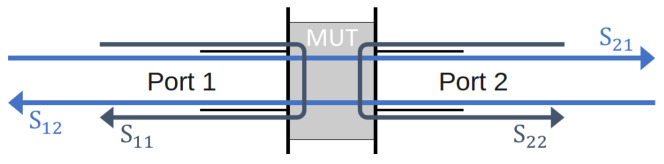
Scheme of the S-parameters in 2-port configuration.

**Figure 2 micromachines-13-02152-f002:**
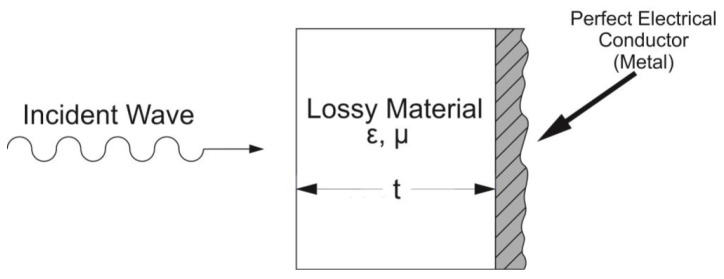
Schematic of the Rozanov configuration: the metallic plane (perfect electrical conductor) reflects the wave transmitted in the lossy slab (white) [26].

**Figure 3 micromachines-13-02152-f003:**
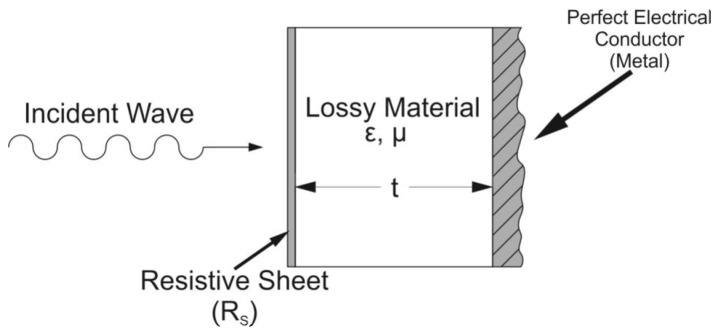
Schematic of the Salisbury configuration: the metallic plane (perfect electrical conductor) reflects the wave transmitted in the lossy slab (white) and strapped thanks to the resistive sheet [26]. The lossy material is set with a thickness of $\frac{\lambda }{4}$ of the aimed frequency.

**Figure 4 micromachines-13-02152-f004:**
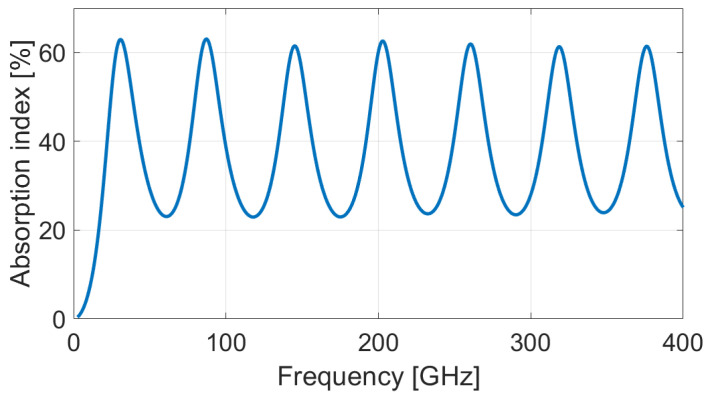
Simulation of absorption index of a PLA-based Salisbury screen. The resistive sheet is a 200 µm-thick rPLA–4%OMMT films and the lossy material is a 1200 µm-thick rPLA film.

**Figure 5 micromachines-13-02152-f005:**
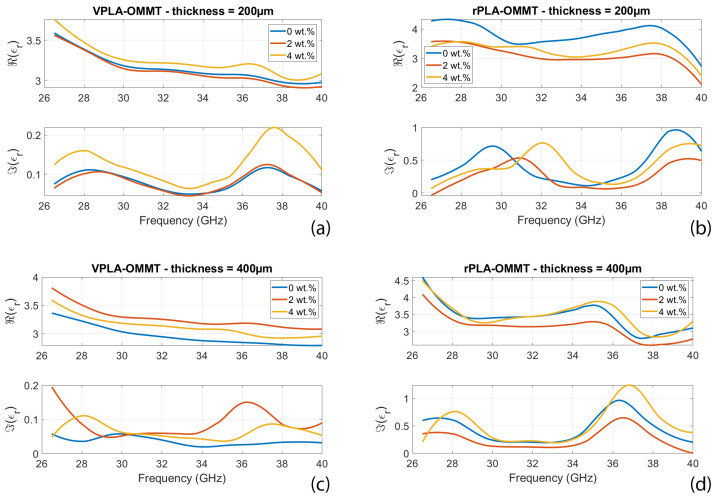
Real and imaginary parts of permittivity versus frequency and thickness of virgin and recycled PLA having various wt.% concentration in OMMT. (**a**) virgin PLA 200 µm-thick, (**b**) recycled PLA 200 µm-thick, (**c**) virgin PLA 400 µm-thick, and (**d**) recycled PLA 400 µm-thick.

**Figure 6 micromachines-13-02152-f006:**
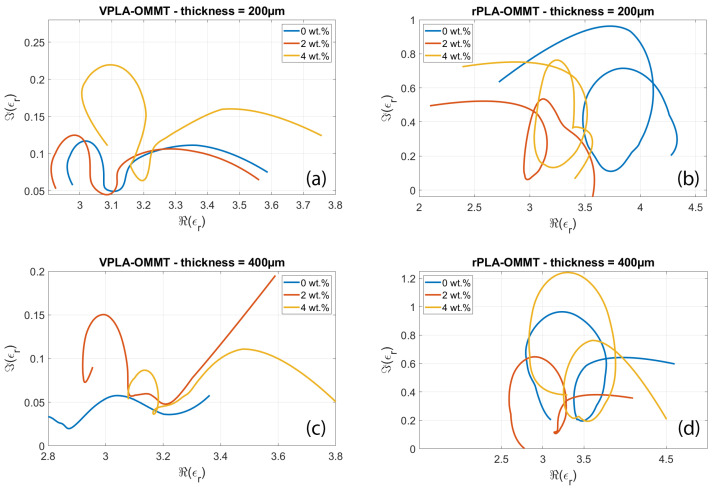
Cole-Cole plot of permittivity for virgin and recycled PLA having various wt.% concentration in OMMT. (**a**) virgin PLA 200 µm-thick; (**b**) recycled PLA 200 µm-thick; (**c**) virgin PLA 400 µm-thick; and (**d**) recycled PLA 400 µm-thick.

**Figure 7 micromachines-13-02152-f007:**
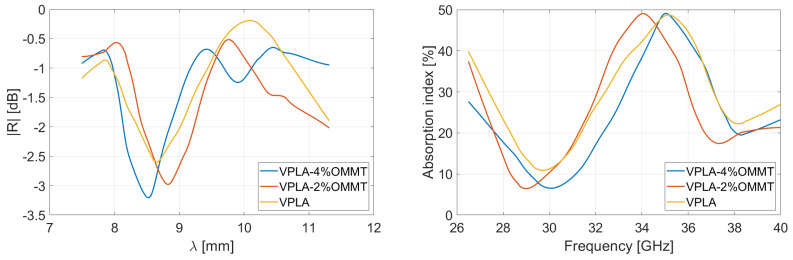
Measurement in Rozanov configuration of reflectivity versus wavelength and absorption versus frequency of virgin PLA films having various wt.% concentration in OMMT. Thickness of films is 400 µm.

**Figure 8 micromachines-13-02152-f008:**
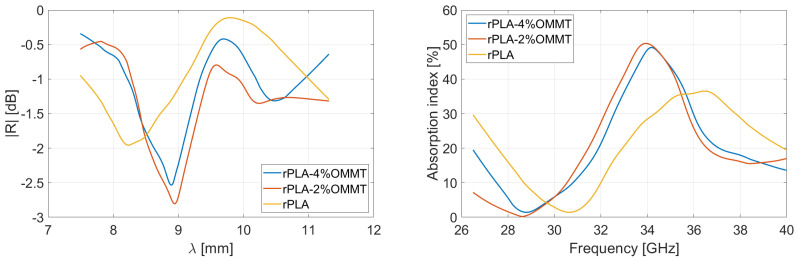
Measurement in Rozanov configuration of reflectivity versus wavelength and absorption versus frequency of recycled PLA films having various wt.% concentration in OMMT. Thickness of films is 400 µm.

**Figure 9 micromachines-13-02152-f009:**
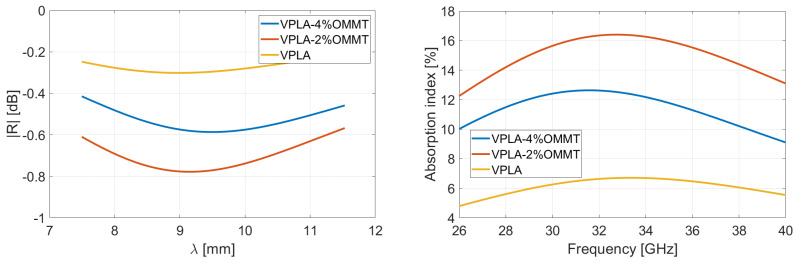
Simulation of reflectivity and absorption performances for virgin PLA–OMMT nanocomposites in Rozanov configuration. The thickness of films is set at 1.4 mm.

**Figure 10 micromachines-13-02152-f010:**
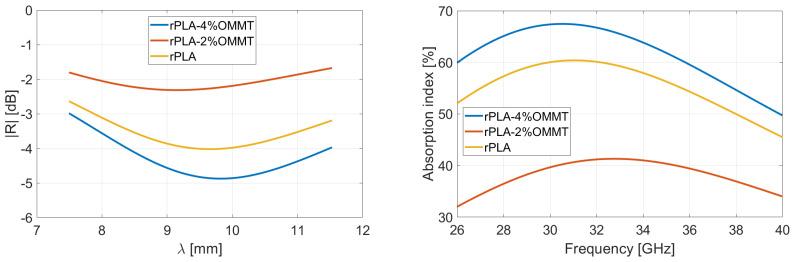
Simulation of reflectivity and absorption performances for recycled PLA–OMMT nanocomposites in Rozanov configuration. The thickness of films is set at 1.4 mm.

**Figure 11 micromachines-13-02152-f011:**
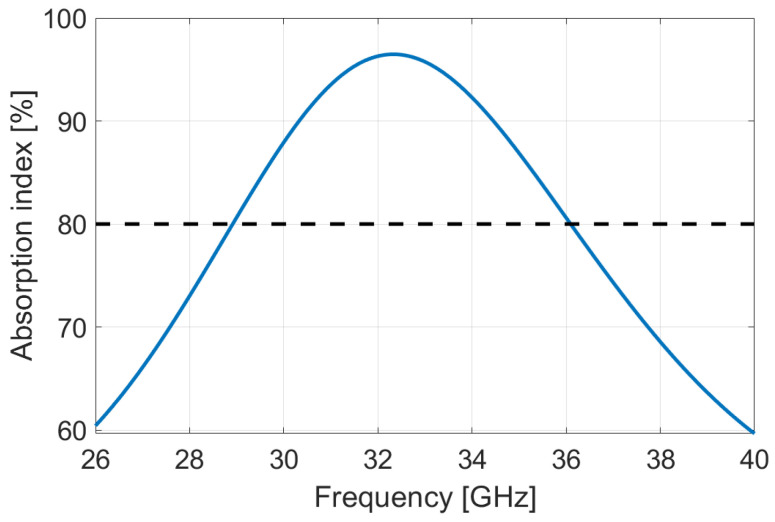
Simulation of absorption of rPLA–4%OMMT nanocomposite in Rozanov configuration, resulting in an optimization on the film thickness in order to reach the average absorption of 80%. The dotted line shows the target.

**Figure 12 micromachines-13-02152-f012:**
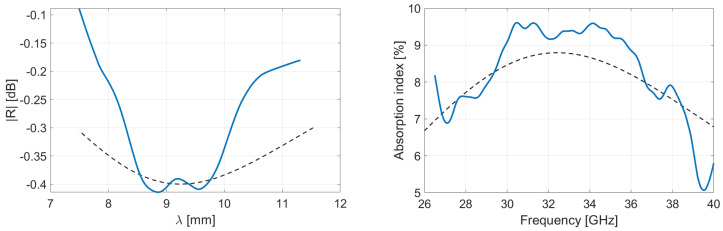
Reflectivity and absorption performances of the virgin Salisbury structure. The resistive layer is VPLA–4%OMMT 200 µm-thick film and the lossy material is VPLA 1.2 mm-thick. The dotted line is the simulation of the performance based on material properties.

**Figure 13 micromachines-13-02152-f013:**
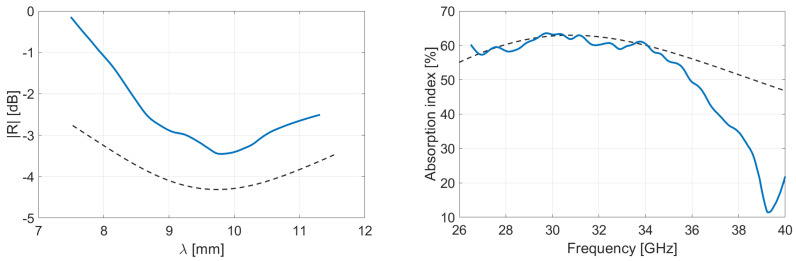
Reflectivity and absorption performances of the recycled Salisbury structure. The resistive layer is rPLA–4%OMMT 200 µm-thick film and the lossy material is rPLA 1.2 mm-thick. The dotted line is the simulation of the performance based on material properties.

**Figure 14 micromachines-13-02152-f014:**
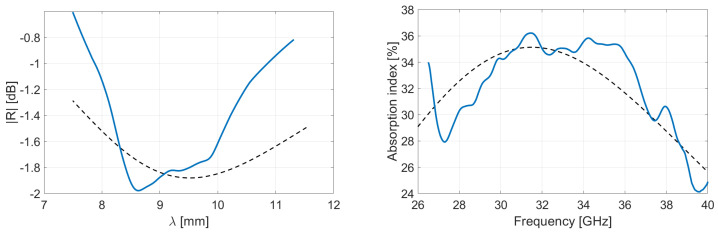
Reflectivity and absorption performances of the hybrid Salisbury structure. The resistive layer is rPLA–4%OMMT 200 µm-thick film and the lossy material is VPLA 1.2 mm-thick. The dotted line is the simulation of the performance based on material properties.

**Table 1 micromachines-13-02152-t001:** Summary of reflectivity and absorption performances for virgin and recycled PLA–OMMT nanocomposites, averaged over the 26.5–40 GHz frequency range.

Sample	Thickness [µm]	Reflectivity [dB]	Absorption [%]	Rozanov Limit [µm]
VPLA	400	−1.40	29.3	31.4
VPLA–2%OMMT	400	−0.98	26.2	33.5
VPLA–4%OMMT	400	−0.99	26.1	31.9
rPLA	400	−1.11	21.8	24.1
rPLA–2%OMMT	400	−0.61	21.5	30.6
rPLA–4%OMMT	400	−0.57	21.1	28.4

**Table 2 micromachines-13-02152-t002:** Summary of reflectivity and absorption performances for simulated virgin and recycled PLA–OMMT nanocomposites, averaged over the 26.5–40 GHz frequency range for film having a thickness fixed at 1.4 mm.

Sample	Thickness [mm]	Reflectivity [dB]	Absorption [%]
VPLA	1.40	−0.28	6.14
VPLA–2%OMMT	1.40	−0.71	15.05
VPLA–4%OMMT	1.40	−0.53	11.42
rPLA	1.40	−3.54	55.62
rPLA–2%OMMT	1.40	−2.10	38.33
rPLA–4%OMMT	1.40	−4.20	61.85

**Table 3 micromachines-13-02152-t003:** Summary Salisbury screens measured under microwave.

Sample	Resistive Layer		Lossy Layer	
	Composition	Thickness	Composition	Thickness
Virgin Salisbury structure	VPLA–4%OMMT	200 µm	VPLA	1.2 mm
Recycled Salisbury structure	rPLA–4%OMMT	200 µm	rPLA	1.2 mm
Hybrid Salisbury structure	rPLA–4%OMMT	200 µm	VPLA	1.2 mm

## Data Availability

The data presented in this study are available upon reasonable request from the corresponding author.

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
