# Peer review of "Absorption Performances of PLA-Montmorillonite Nanocomposites Thin Films in Salisbury and Rozanov Configurations: Influence of Aging and Mechanical Recycling"

_micromachines, 2022, doi:10.3390/mi13122152_

Round 1

Reviewer 1 Report

In this work, nanocomposites combining poly(lactic acid) (PLA) and organo-modified montmorillonite (OMMT) are considered for electromagnetic interferences (EMI) shielding applications. Composites based on virgin as well as recycled PLA after photochemical, thermal and hydrolytic ageing are compared for various film thicknesses and weight concentrations of OMMT in the PLA matrix. Interestingly, the potential of recycling has been demonstrated. I support publication after consideration of the following comments.

-          I suggest to very briefly mention the difference between the Rozanov and Salisbury configuration in the abstract for a general reader.

-          A better framing of why PLA is considered is needed in the introduction. Also a better framing of extrusion-based circular solutions is recommended.

-          Section 2.1: the details given on the melt extrusion and compression molding steps are quite limited.

-          The aspect of ageing of virgin PLA films vs. aging of virgin PLLA-OMMT nanocomposite films (it’s explained differently in the abstract and Section 2.1) should be made consistent.

-          An indication of the actual times corresponding to the accelerated ageing conditions would be useful. Also, what are the conditions applied during photochemical ageing?

-          Figure 2 is slightly cut off.

-          An illustration of the meaning of S11, S22, S21 and S12 would be helpful for a general reader.

-          Please carefully check the text with respect to typos and the construction of certain sentences. Some examples:

-          “Various MAS has been studiedà “Various MAS have been studied”

-          Most of researchà “Most research”

-          aiming to reduce the world’s plastic growing wasteà “aiming to reduce the world’s growing amount of plastic waste”

-          through the measurement of R since if R is minimized. It means that …” à should be rewritten

-          Line 118 should be rewritten

Reviewer 2 Report

1. How did the authors fabricate the PLA-OMMT nano composites into the absorbing structures by integrating the Salisbury sheet? The fabrication process should be described clearly.

2. How did the authors carry out the optimization to the structure as the size of the structure was important to the effects of absorption? The explanation should be given.

3. Please explain the Cole-Cole pattern in Fig. 5 and give reasons why the electromagnetic properties exhibited several cycles.

4. Please explain the absorption peaks in Fig. 6 and 7 for more details. Did the absorption peak frequency relate to the thickness of the structure?

5. To present more comprehensive explanation for the meaning of the work, some important previous works should be cited for better description as follows.

[1] Zhang W, Zhao B, Ni N, Xiang H, Dai F-Z, Wu S, et al., High entropy rare earth hexaborides/tetraborides (HE REB6/HE REB4) composite powders with enhanced electromagnetic wave absorption performance, J. Mater. Sci. Technol., 87 (2021) 155-166.

[2] An Z, Li Y, Luo X, Huang Y, Zhang R, Fang D, Multilaminate metastructure for high-temperature radar-infrared bi-stealth: Topological optimization and near-room-temperature synthesis, Matter, 5 (6) (2022) 1937-1952.

[3] Zhan Y, Xia L, Yang H, Zhou N, Ma G, Zhang T, et al., Tunable electromagnetic wave absorbing properties of carbon nanotubes/carbon fiber composites synthesized directly and rapidly via an innovative induction heating technique, Carbon, 175 (2021) 101-111.

[4] Song Q, Ye F, Kong L, Shen Q, Han L, Feng L, et al., Graphene and MXene Nanomaterials: Toward High-Performance Electromagnetic Wave Absorption in Gigahertz Band Range, Adv. Funct. Mater., (2020) 2000475.

[5] Zhou N, Zhang L, Wang W, Zhang X, Zhang K, Chen M, et al., Stereolithographically 3D Printed SiC Metastructure for Ultrabroadband and High Temperature Microwave Absorption, Advanced Materials Technologies, (2022) 2201222.
